# Research on Receiving Seeds Performance of Belt-Type High-Speed Corn Seed Guiding Device Based on Discrete Element Method

Chengcheng Ma , Shujuan Yi *, Guixiang Tao, Yifei Li, Song Wang, Guangyu Wang and Feng Gao

College of Engineering, Heilongjiang Bayi Agricultural University, Daqing 163319, China; mcc1995@protonmail.com (C.M.)
* Correspondence: yishujuan_2005@126.com; Tel.: +86-138-3696-1877

**Abstract:** Because the initial speed of the seeds leaving the seed disk is too high, they collide and bounce off the inner wall of the seed guide tube, resulting in poor sowing quality when corn is sown at high speeds above 12 km/h. This study clarifies the primary factors affecting the stability of seed receiving and the accuracy of the seed entering the seed cavity, establishes the dynamic model of seed clamping, transportation, and releasing, and investigates the belt-type high-speed corn seed guiding device with the seed receiving system as the research object. It also proposes an improved method of adding herringbone lines on the finger surface to address this issue. Using EDEM software, a virtual experiment of seed-receiving performance was conducted, and the change trend of stress on seeds with and without a herringbone pattern and different wheel center distance as well as the change trend of the speed of seeds with various feeder wheel speeds and finger length, were both examined. The outcomes of the simulation demonstrate that the herringbone-lined feeder wheel could increase the stress on seeds. The average value of the stress on the seeds is the highest at the wheels' center distance of 37 mm. The stability and speed fluctuation of seeds introduced into the seed cavity were better when the feeder wheel speed was 560 r/min. The speed of fluctuation and stability of the seeds introduced into the seed cavity were better when the finger length was 12 mm. The high-speed camera test on the test bench was used to verify the seed guiding process in accordance with the simulation results, and the outcomes were largely consistent. The study's findings can serve as a theoretical foundation for a belt-type high-speed corn seed guiding device optimization test.

**Keywords:** corn; high-speed sowing; seed receiving system; test



## 1. Introduction

One of the most popular cereal crops in the world and a staple food for millions of people is corn (*Zea mays* L.) [1]. Corn is a crop that is grown for human consumption as well as for animal feed, the production of biofuel, and industrial uses. Corn planting requires both high quality and high speed because corn has a large planting area, a short planting period, and the quality of the planting directly affects the yield [2]. A typical seed guide tube is designed for low-speed operation and has a straightforward structure. A serious collision between the seeds and the seed guide tube's inner wall occurs when seeds are sown at high speeds, which reduces the seed placement accuracy and leads to uneven sowing. The belt-type seed guide device uses synchronous seed transportation by the seed conveying belt, which reduces violent collisions during seed transportation, ensures that seeds are discharged from the seed metering device in a uniform and orderly state, and can meet the demands of precision seeders operating at high speeds [3,4]. A belt-type high-speed corn seed guiding device with a seed receiving system created by the Precision Planting Company can actively restrain the corn seeds as they enter the seed cavity from the seed disk, improving their stability and enabling high-speed, precise sowing at a speed of 16 km/h. The feeder wheel's seed-receiving mechanism, however, has not been the

subject of any research, and it is unclear what variables influence how well it receives seeds. Therefore, this study investigated the seed receiving mechanism of the seed guiding device, enhanced key components, and carried out related experiments, which will be crucial for high-speed corn sowing in the future.

The discrete element method (DEM), one of many methods for researching corn sowing technology, has drawn a lot of attention lately. DEM-based simulations can provide insights into the interaction between the seed particles and planting equipment including the trajectory, velocity, and impact force of the seeds, which are difficult to obtain through traditional experimental methods [5,6]. Therefore, DEM-based simulations have been used to investigate various aspects of corn seed planting such as seed spacing, seed depth, and seed orientation [7,8].

Several studies have utilized EDEM software to simulate and optimize the seed placement process in precision planters. For instance, Liu et al. used EDEM to investigate the effect of seed size and shape on the seed flow characteristics in a planter with a seed disk metering device [9]. They found that the seed size and shape significantly influenced the seed flow rate and uniformity. Similarly, Wang et al. used EDEM to simulate the seed dropping process in a pneumatic planter with a curved seed tube [10]. They found that the curved seed tube could significantly improve the seeding accuracy. In another study, Zou et al. used EDEM to optimize the seed metering mechanism in a precision planter. They found that the seed metering device's design significantly influenced the seed metering accuracy and uniformity [11]. Li et al. also used EDEM to investigate the effect of seed size and shape on the seed flow characteristics in a vertical drum [12]. They found that the drum's rotating speed significantly influenced the seed flow rate and uniformity.

Several other studies have also utilized EDEM software to optimize various aspects of the seed placement process in precision planters. For instance, Fan et al. used EDEM and response surface methodology to optimize the seed-metering device and seed distribution system in a precision planter [13]. Similarly, Huang et al. used EDEM to investigate the effect of the seed size and shape on the seed metering accuracy in a vacuum planter [14], while Zhang et al. used EDEM to investigate the effect of the seed size and shape on the seed metering accuracy in a disc planter [15].

In conclusion, EDEM software has become a potent tool for modeling and perfecting the seed placement procedure in precision planters. As a result, the dynamic model of the seed receiving process was established in this study, the research carrier was a belt-type high-speed corn seed guiding device, and the primary factors affecting seed receiving performance were determined. Based on this, the process of corn seeds being incorporated into the seed receiving system was examined using EDEM simulation technology, with a focus on the force of the feeder wheel on corn seeds and the variation law of the speed of seeds entering the seed cavity. Bench tests were used to confirm the validity of the simulation results.

## 2. Materials and Methods

### 2.1. Structure and Operating Principle of Belt-Type Corn Seed Guiding Device

#### 2.1.1. Device Structure

Figure 1 depicts the structure's overall structure. The main parts of the entire apparatus are the seed protection cover, seed guiding belt shell, feeder wheel, seed throwing plate, pre-tightening spring, seed cleaning claw, driving pulley, driven pulley, motor, and seed cleaning claw. Three steps make up its working process: receiving seeds, transporting seeds, and throwing seeds.

#### 2.1.2. Principle of Operation

The seed-guiding device is attached to the seed discharge port of the seed metering device. The primary feeder wheel and the auxiliary feeder wheel transport and discharge the seeds to the seed cavity of the conveying belt after rotating and clamping them before they leave the seed disk. The average working speed can reach 16 km/h as the seeds are

moved to the seed throwing location under the rotation of the seed conveying belt and deposited into the seed ditch [16–19].

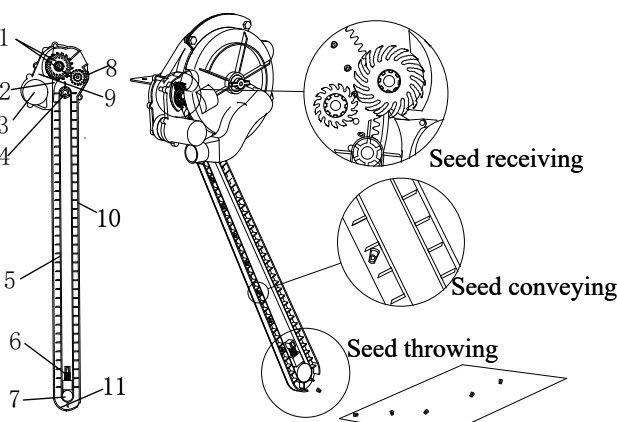

**Figure 1.** Structure diagram of the belt-type high-speed corn seed guiding device. 1. Feeder wheel. 2. Seed cleaning claw. 3. Motor. 4. Driving pulley. 5. Seed conveying belt. 6. Pre-tightening spring. 7. Driven pulley. 8. Gear box. 9. Seed protection cover. 10. Seed conveying belt shell. 11. Seed throwing plate.

### 2.2. Analysis of Seed Receiving System

2.2.1. Composition and Operating Principle of Seed Receiving System

The seed receiving system of the belt-type high-speed corn seed guiding device is primarily composed of the components shown in Figure 2: a primary feeder wheel, an auxiliary feeder wheel, a seed cleaning claw, a gear shaft, a gear box, a motor, and a rear cover. Among them, the main and auxiliary feeder wheels are the essential elements of the seed receiving system, and the stability and precision of the seed guiding device directly affects how well they function. The rubber feeder wheel has fingers with an inclined arc structure that are evenly spaced across its outer surface. The primary feeder wheel has 17 fingers, while the auxiliary feeder wheel has 15 fingers. Below the main feeder wheel is a seed cleaning claw, and the claw tip and finger are spaced apart to prevent seeds from getting caught between the fingers and interfering with the seed receiving process.

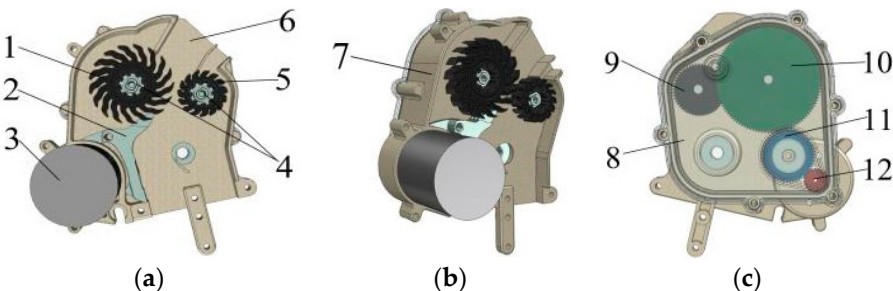

(**a**)　　　　　　　　　　(**b**)　　　　　　　　　　(**c**)

**Figure 2.** Schematic diagram of seed receiving system. 1. Primary feeder wheel. 2. Seed cleaning claw. 3. Motor. 4. Gear shaft. 5. Auxiliary feeder wheel. 6. Seed receiving port. 7. Gear box. 8. Rear cover. 9. Driving auxiliary feeder wheel gear. 10. Driving primary feeder wheel gear (dual gear). 11. Carrier gear. 12. Motor gear. (**a**) Front view; (**b**) Side view; (**c**) Back view.

As shown in Figure 3, there are three stages to the seed receiving system's operation process: clamping seeds (I), transporting seeds (II), and releasing seeds (III). Where $v_0$ is the forward sowing speed in m/s; $\omega_1$ is the primary feeder wheel's rotational angular velocity in rad/s; and $\omega_2$ is the auxiliary feeder wheel's rotational angular velocity in rad/s, $\omega_2 = 1.7 \cdot \omega_1$.

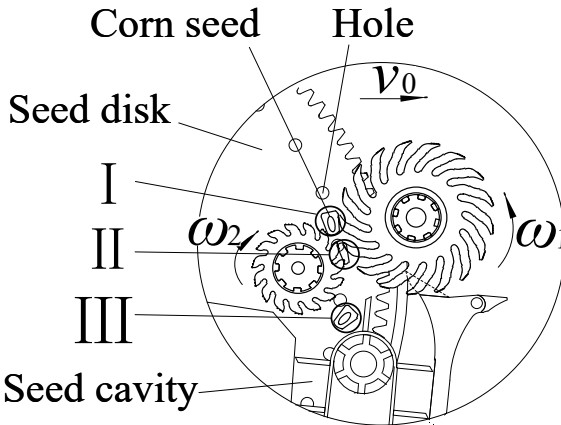

**Figure 3.** Operating process diagram of the seed receiving system.

The motor causes the primary feeder wheel and the auxiliary feeder wheel to rotate in opposition during the sowing process. The primary feeder wheel and the auxiliary feeder wheel "pick" the seed by rotating and clamping, and this working process is clamping the seeds, when the seed disk transports seeds to the receiving port. Transporting the seeds occurs as a result of the clamped seeds moving beneath the primary and auxiliary feeder wheels' clamping. The seeds are then released by the feeder wheel into the seed cavity of the seed conveying belt during this working process. The seeds can be steadily and evenly moved from the seed metering device to the seed cavity of the seed conveying belt by using this technique of "picking" and then "releasing" the seeds.

2.2.2. Mechanical Analysis of Seeds Receiving Process

The dynamics of seeds in three stages—clamping, transporting, and releasing by the feeder wheel—were examined in order to study the stability of the seed receiving system and the accuracy of seeds entering the seed cavity.

(1) Mechanical analysis of clamping seeds

The primary feeder wheel and secondary feeder wheel simultaneously apply friction forces $f_1$ and $f_2$ to the seeds when the feeder wheel clamps them. The combined force of $f_1$ and $f_2$ is what makes up the clamping force $F_c$ of the feeder wheel on the seeds. As shown in Figure 4, the seeds are additionally subject to the forces of gravity $G$, centrifugal force $J$, suction force $P$ of negative pressure on the seeds, and supporting force $N$ of the hole on the seeds.

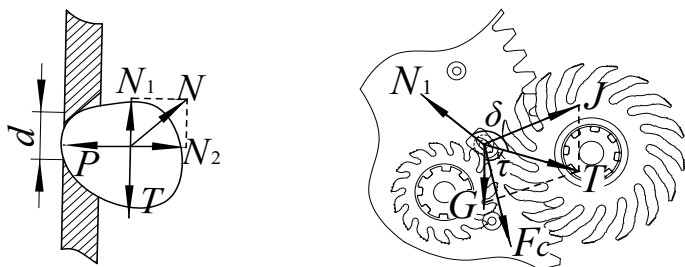

**Figure 4.** Schematic diagram of the seed clamping process.

The clamping force $F_c$ of the feeder wheel on the seeds should be greater than $T$ (the resultant force of seed gravity $G$ and centrifugal force $J$) and $N_1$ (the supporting force of the hole on the seeds in the direction parallel to the seed disk) if the seeds are removed smoothly.

$$F_c > \sqrt{T^2 + N_1^2 + 2TN_1 \cos \delta} \tag{1}$$

where $F_c$ is the clamping force of feeder wheel on seeds; $T$ is the resultant force of gravity and inertia; $N_1$ is the supporting force of the hole to the seeds along the direction parallel to the seed disk; $\delta$ is the angle between $T$ and $N_1$.

The combined force $T$ of the seed gravity, centrifugal force, and the hole's supporting force $N_1$ acting in a direction parallel to the seed-metering device can be written as follows:

$$\begin{cases} T = J \cdot G \sin \tau \\ N_1 = N_c \cdot P \end{cases} \tag{2}$$

Combining Formulas (1) and (2) results in:

$$F_c > \sqrt{J^2 G^2 \sin^2 \tau + Nc^2 P^2 + 2JG \sin \tau \cdot NcP} \tag{3}$$

They include:

$$\begin{cases} J = 4\pi^2 m D n_1^2 \\ P = (\pi d^2 / 4)(P_a - P_0) \end{cases} \tag{4}$$

where $m$ is the seed quality; $D$ is the distance from the center of gravity of corn seed to the center of hole; $n_1$ is the rotating speed of seed disk; $P$ is the suction force on seeds; $Pa$ is the atmospheric pressure; $P_0$ is the vacuum chamber pressure; $D$ is the diameter of hole; $N_c$ is the coefficient of supporting force; $N$ is the hole's support for seeds; $J$ is the centrifugal force of seed disk on seeds; $G$ is the seed gravity; $\tau$ is the angle between $G$ and $J$.

The centrifugal force of the seed disk and the suction force of the fan on the seeds must be overcome for the feeder wheel to clamp the seeds, according to Formulas (3) and (4). In order to bring the seed clamping on the hole into the seed cavity of the seed conveying belt below, the feeder wheel must have sufficient clamping force on the seeds (i.e., to increase the friction between the finger and the seeds).

(2)    Mechanical analysis of transporting seeds

The seeds will be subjected to the clamping force $F_c$ (the sum of $f_1$ and $f_2$) of the feeder wheels, the supporting forces $F_{N1}$ and $F_{N2}$ of the primary and secondary feeder wheels, and the gravity $G$ of the seeds themselves during the transportation stage after the primary and secondary feeder wheels have taken the seeds down. Establish the rectangular coordinate system *xoy* using the seed center as the coordinate origin. The force acting on the seed at this time is depicted in Figure 5.

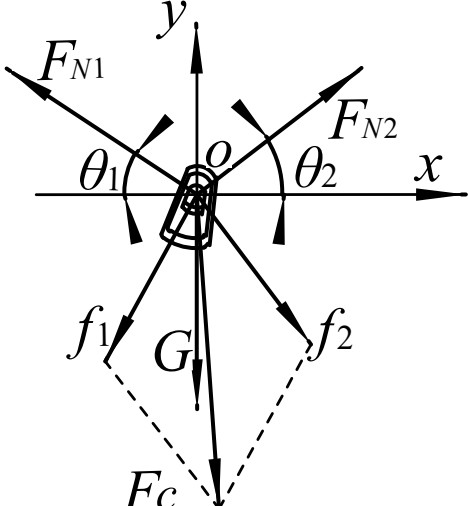

**Figure 5.** Force diagram of the seed during clamping.

The primary and auxiliary seed feeder wheels' $F_{N1}$ and $F_{N2}$ supporting forces are expressed as follows:

$$\begin{cases} F_{N1} = \frac{G}{\sin\theta_1} \\ F_{N2} = \frac{G}{\sin\theta_2} \end{cases} \tag{5}$$

They include:

$$\begin{cases} f_1 = \mu F_{N1} \\ f_2 = \mu F_{N2} \end{cases} \tag{6}$$

The clamping force $F_c$ is the resultant force of the friction forces $f_1$ and $f_2$ of the primary and auxiliary feeder wheels on the seeds at a specific time point:

$$F_c = \sqrt{f_1^2 + f_2^2 + 2f_1 f_2 \cos(\theta_1 + \theta_2)} \tag{7}$$

where $F_{N1}$ is the supporting force of the fingers of the primary feeder wheel on the seeds; $F_{N2}$ is the supporting force of the fingers of the auxiliary feeder wheel on the seeds; $\theta_1$ is the included angle between the seed supporting force of the primary feeder wheel and the horizontal plane; $\theta_2$ is the included angle between the seed supporting force of the auxiliary feeder wheel and the horizontal plane; $\mu$ is the friction coefficient between finger and seed; $f_1$ is the friction of the primary feeder wheel against the seed; $f_2$ is the friction of the auxiliary feeder wheel against the seed.

The following can be attained when Equations (5)–(7) are combined:

$$F_c = \mu \cdot G \sqrt{\left(\frac{1}{\sin^2\theta_1} + \frac{1}{\sin^2\theta_2} + \frac{2\cos(\theta_1 + \theta_2)}{\sin\theta_1 \cdot \sin\theta_2}\right)} \tag{8}$$

The included angles $\theta_1$, $\theta_2$ between the feeder wheel's supporting force on the seeds and the horizontal plane and the friction coefficient have an impact on the clamping force $F_c$ on the seeds when the feeder wheel transports the seeds, as shown by Formula (8). The friction between the fingers and seeds can be improved by raising the coefficient of friction between the two, which also prevents slipping. As seen in Figure 6, the value of the included angle between the feeder wheel's seed-supporting force and the horizontal plane is correlated with the wheels' center distance $L$. $\theta_1$ and $\theta_2$ increase with the decrease in the wheels' center distance $L$ ($\theta_1 > \theta_1'$, $\theta_2 > \theta_2'$), and the supporting force of the feeder wheel on the seed increases accordingly ($F_{N1} > F_{N1}'$, $F_{N2} > F_{N2}'$), which improves the friction between the finger and the seed and avoids the sliding phenomenon between the fingers and the seeds. The seed receiving load will increase and the finger wear will worsen if the wheels' center distance is too small. The finger and the seed will slide relatively and lose the active seed receiving function if the wheels' centers are too far apart. Therefore, this study conducted simulation test research in order to further clarify the impact of the wheels' center distance on seed stress.

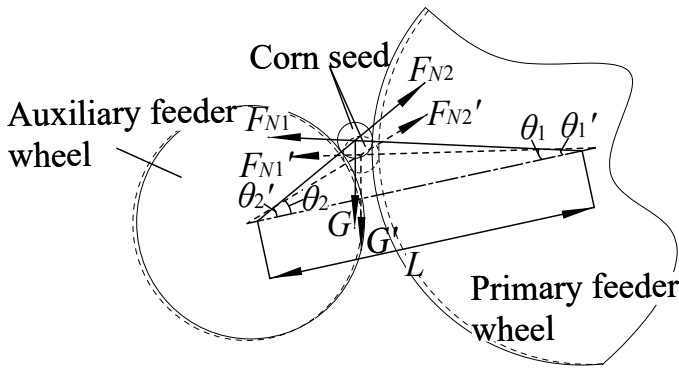

**Figure 6.** Schematic diagram of the change of supporting force with the wheels' center distance.

(3)   Mechanical analysis of releasing seeds

The feeder wheel moves the seeds, which are then released into the conveying belt's seed cavity. Under the influence of the seed gravity $G$ and the elastic forces $F_{t1}$ and $F_{t2}$ of the finger on the seeds, when the seeds leave the feeder wheel, they will be thrown obliquely into the seed cavity (Figure 7), where Fe is the combined force of the elastic force Ft of the finger on the seed and the seed gravity.

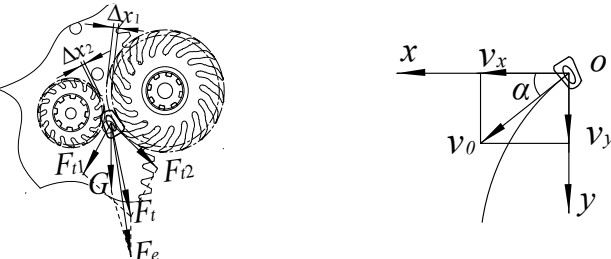

**Figure 7.** Schematic diagram of the force and movement of the releasing seeds.

The seeds' own gravity and the elastic force the fingers exert on them when they are returned to their initial state are as follows:

$$\begin{cases} F_{t1} = k\Delta x_1 \\ F_{t2} = k\Delta x_2 \\ G = mg \end{cases} \tag{9}$$

The acceleration of the seed falling off the feeder wheel is as follows:

$$a = \frac{F_e}{m} \tag{10}$$

where $F_e$ is the resultant force of the elastic force of fingers on seeds and the gravity of seeds; $F_{t1}$ is the elastic force of the fingers of the primary feeder wheel on the seeds; $F_{t2}$ is the elastic force of the fingers of the auxiliary feeder wheel to the seeds; $\Delta x_1$ is the finger shape variable of primary feeder wheel; $\Delta x_2$ is the fingering shape variable of the auxiliary feeder wheel; $k$ is the elastic coefficient of fingers; $g$ is the acceleration of gravity.

Following an oblique downward throwing motion along the x and y axes, the speed at which the seeds enter the seed cavity is as follows:

$$\begin{cases} v_x = v_0 \cos \alpha \\ v_y = v_0 \sin \alpha + at \end{cases} \tag{11}$$

They include:

$$v_0 = 2\pi R n_2 \tag{12}$$

The simultaneous Formulas (10)~(12) can be obtained as follows:

$$\begin{cases} v_x = 2\pi R n_2 \cdot \cos \alpha \\ v_y = 2\pi R n_2 \cdot \sin \alpha + F_e \cdot t/m \end{cases} \tag{13}$$

where $v_x$ is the speed of seeds on the $x$ axis; $v_y$ is the speed of seeds in the direction of the y axis; $R$ is the feeder wheel radius; $n_2$ is the feeder wheel speed; $v_0$ is the initial speed of oblique downward throwing of seeds; $\alpha$ is the angle at which seeds are thrown obliquely downward; $t$ is the time taken for the seed to enter the seed cavity; $a$ is the seed released acceleration.

According to Formula (13), the radius of the feeder wheel, the finger shape variable, and the feeder wheel speed are all related to how quickly seeds enter the seed cavity. The finger length and the properties of its material are related to the finger's shape. The longer the finger, assuming the processing material remains constant, the greater the deformation

and elasticity will be when it comes into contact with the seed, causing the seed to enter the seed cavity more quickly. The risk of seed jam between the fingers will, however, increase with the long finger. The initial speed of oblique downward throwing and the speed of entering the seed cavity all increase with the feeder wheel speed. The fingers will, however, collide against the seed cavity's inner wall if it is too long, which will decrease their accuracy as they enter the cavity. The flexible fingers that make up the feeder wheel's circumference cause the circumferential radius to fluctuate uncontrollably when the fingers come into contact with the seeds. This study conducted a simulation test to investigate the impact of the finger length and the feeder wheel speed of the seeds entering the seed cavity in order to increase the operability and accuracy of the simulation and bench test.

### 2.3. Improvement Scheme of the Feeder Wheel

According to the above analysis of the strain placed on the seeds by the feeder wheel during each of the three stages of clamping, transporting, and releasing, increasing the friction coefficient between the feeder wheel and the seeds can significantly improve the clamping force of the feeder wheel on the seeds. Therefore, in this study, herringbone lines were added to the surface of the feeder wheel by taking advantage of the characteristics of a large friction coefficient, good anti-skid performance, and guidance (Figure 8). The primary and auxiliary feeder wheels' center distances and the addition of herringbone lines can be known to have an effect on the seed receiving effect, while the fingering variable and the feeder wheel speed are known to have an effect on the speed at which seeds enter the seed cavity. In order to investigate the effects of adding herringbone lines, the wheels' center distance, finger length, and feeder wheel speed on the force and speed of the rotating corn seeds entering the seed cavity, this study conducted simulation experiments using EDEM software.

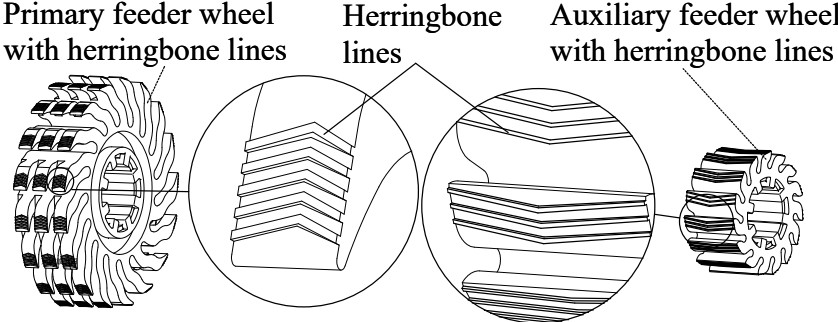

**Figure 8.** Primary and auxiliary feeder wheel with herringbone lines added.

### 2.4. EDEM Simulation and Analysis of Seed Receiving

2.4.1. Particle Modeling of Corn Seeds

Currently, a multi-sphere model is primarily used in the EDEM simulation of irregularly shaped particles to produce a surface group that more closely matches the outer contour of the simulation particles [20–24]. The particles produced using this technique are made up of numerous spheres and are recognized as independent particles by EDEM [25–28].

One hundred "Demeiya No. 1" corn seeds were used in this study as the research object, and their triaxial dimensions were measured. The outcomes are displayed in Table 1. Table 1 shows that the "Demeiya No. 1" corn seeds had a length dimension that is greater than their width and thickness dimensions, and that the length dimension difference was greater than the width and thickness dimension difference. The triaxial size of the corn seeds was subjected to a KS test, and the results revealed that its significant length, width, and thickness indices were all greater than 0.05, suggesting that it roughly obeys the normal distribution.

**Table 1.** Triaxial dimensions of corn seeds.

| Index | Minimal Value/mm | Maximum Value/mm | Mean Value/mm | Variance |
|---|---|---|---|---|
| Length | 9.89 | 12.08 | 11.564 | 0.133 |
| Width | 6.07 | 8.61 | 8.212 | 0.076 |
| Thickness | 4.01 | 5.98 | 4.23 | 0.017 |

According to their external dimensions, the corn seeds were divided into three grades: big flat, medium flat, and small flat. A German GOM ATOS Core-mv185 scanner was used to scan the corn seeds, and the resulting three-dimensional model of the corn seeds was saved in STL format and imported into EDEM. Figure 9 shows the physical diagrams, diagrams from a three-dimensional scanning model, and diagrams from a particle model filling of corn seeds.

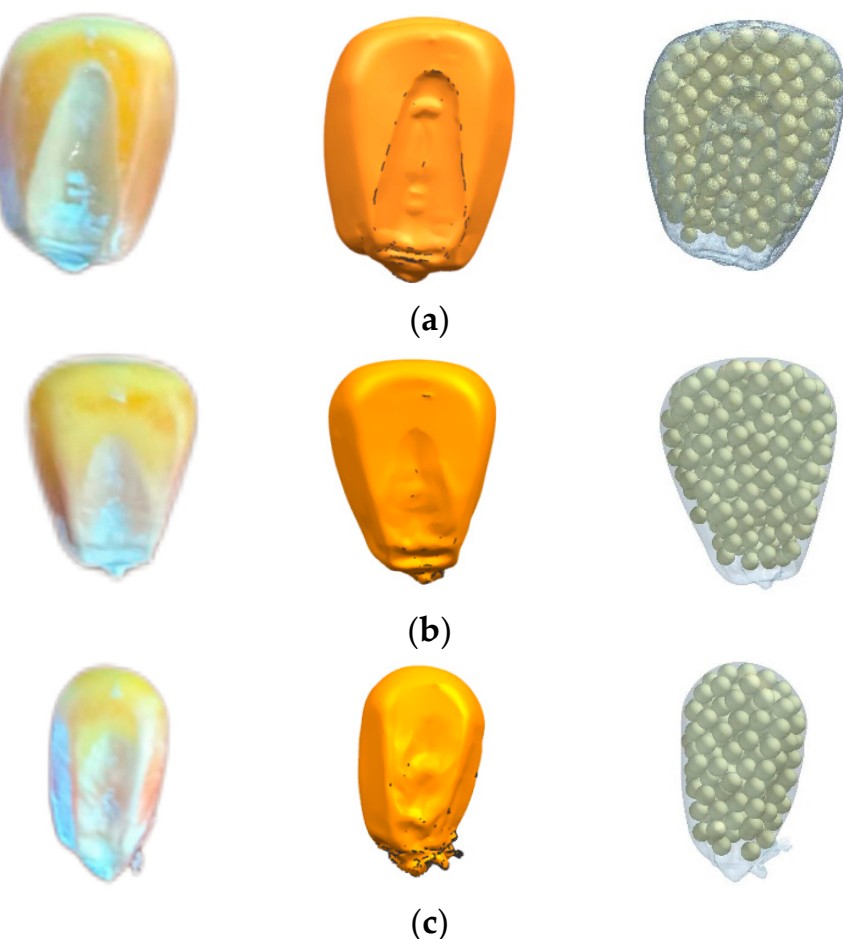

(**a**)

(**b**)

(**c**)

**Figure 9.** Simulation models of the corn seeds. (**a**) Big flat; (**b**) Medium flat; (**c**) Small flat.

2.4.2. Geometric Modeling

The geometric model is the component of the seed metering device and receiving system that the particles in the simulation come into contact with [29–33]. The corn seeds, the seed disk, the primary and secondary feeder wheels, the driving pulley, and the seed conveying belt are the six parts that make up the simplified model of the seed metering device and seed receiving system used in this study. After creating the geometric model in UG, it is exported as a STL file and imported into EDEM. Figure 10 depicts the simulation model for the seed metering device and seed receiving system in EDEM.

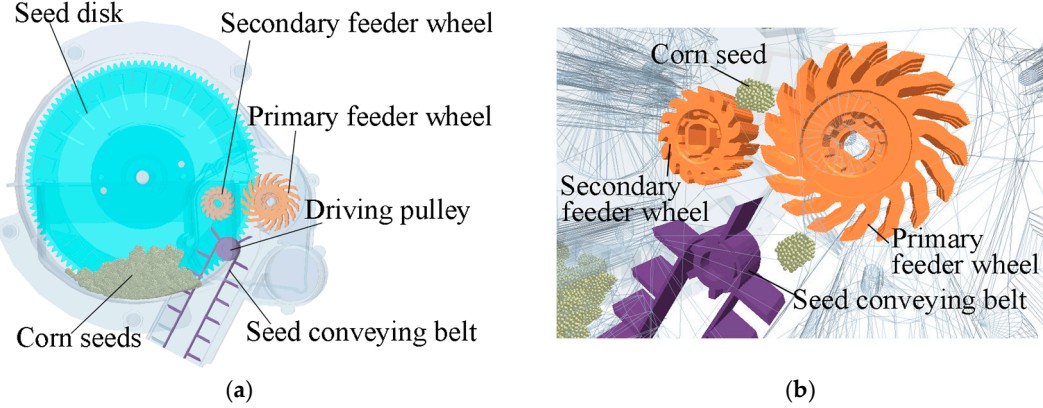

**Figure 10.** Simulation model of receiving seeds. (**a**) Simulation model; (**b**) State of receiving seeds.

2.4.3. Parameter Setting of the Simulation

The seed disk is made of ABS plastic, the seed conveying belt made of polyurethane, and the feeder wheel of the seed receiving system was made of rubber. The physical properties displayed in Table 2 after examining the relevant literature [34,35]. According to the percentage of mixed seeds in the granule factory, 200 medium flat seeds, 100 large flat seeds, and 50 small flat seeds were produced. The percentage of Rayleigh time step was 15%, the fixed time step was $1.95 \times 10^{-2}$ ms, and the simulation time was 10 s in order to guarantee the simulation continuity.

**Table 2.** Physical characteristics of the materials.

| Project | Poisson's Ratio | Shear Modulus/Pa | Density/g·cm$^{-1}$ |
|---|---|---|---|
| Corn seed | 0.40 | $1.77 \times 108$ | 1.180 |
| ABS plastic | 0.50 | $1.37 \times 108$ | 1.197 |
| Rubber | 0.47 | $2.90 \times 109$ | 0.940 |
| Polyurethane | 0.42 | $3.77 \times 107$ | 1.650 |

The contact characteristics between the materials are displayed in Table 3.

**Table 3.** Contact characteristics between the materials.

| Project | Elastic Recovery Coefficient | Coefficient of Sliding Friction | Coefficient of Rolling Friction |
|---|---|---|---|
| Corn seed-corn seed | 0.182 | 0.431 | 0.0782 |
| Corn seed-ABS plastic | 0.621 | 0.482 | 0.0931 |
| Corn seed-rubber | 0.134 | 0.867 | 0.8143 |
| Corn seed-polyurethane | 0.122 | 0.468 | 0.0950 |

## 3. Results and Discussion

### 3.1. Corn Seed Stress Analysis

3.1.1. Stress Analysis of Corn Seeds with Herringbone Lines

The friction coefficient of the finger on the seeds can be directly increased by adding herringbone lines to the surface of the feeder wheel. Comparative simulation experiments between feeder wheels with herringbone lines and feeder wheels without herringbone lines were conducted to investigate the impact of feeder wheels on the clamping force of corn seeds. The stress of corn seeds was used as an index to evaluate the clamping force of feeder wheels on the corn seeds. We set the finger length to 12 mm, the feeder wheel speed to 560 r/min, and the wheels' center distance to 37 mm. Figure 11 depicts the line graph of the stress on seeds over time.

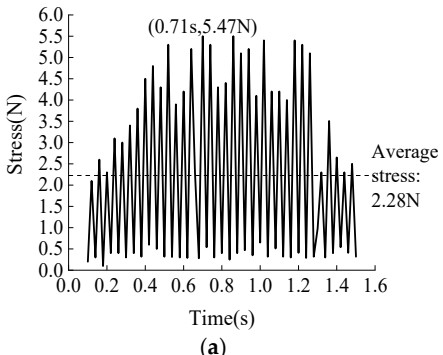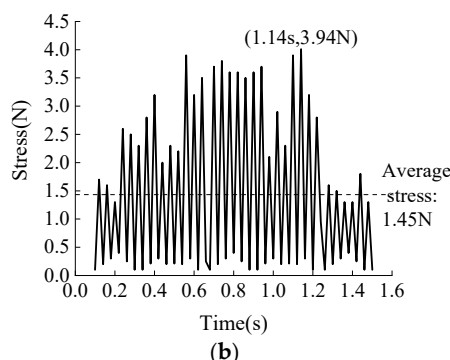

**Figure 11.** Effects of the feeder wheel with and without herringbone lines on the stress of seeds. (**a**) Line graph of the stress change of corn seeds with herringbone lines. (**b**) Line graph of the stress change of corn seeds without herringbone lines.

Figure 11 illustrates how the stress on corn seeds varies erratically over time. The stress on corn seeds is relatively high when the feeder wheel with herringbone lines is used (Figure 11a), with an average stress of 2.28 N and a peak stress of 5.47 N at 0.71 s. The clamping force on the seeds increased with the stress value, placing the seeds in an advantageous clamping state. The stress on corn seeds was relatively low when there were no herringbone lines on the feeder wheel (Figure 11b), with an average stress of 1.45 N and a peak value of 3.94 N at 1.14 s. Through numerical comparison, it can be seen that the average and peak values of the stress on corn seeds produced by the feeder wheel with herringbone lines were higher than those produced by the feeder wheel without herringbone lines. This indicates that the herringbone-lined feeder wheel will increase the clamping force on seeds and significantly enhance the clamping performance on seeds.

### 3.1.2. Force Analysis of Wheels Center Distance on Corn Seeds

The clamping force of the feeder wheels on corn seeds is directly impacted by altering the wheel centers of the primary and auxiliary feeder wheels. In a certain range, the feeder wheels' supporting force on seeds increases as the wheels' center distance decreases, which in turn increases the feeder wheels' clamping force on the seeds. To gauge the clamping force of the feeder wheels on seeds, the stress of the clamped corn seeds can be used. A stronger clamping force is advantageous for feeder wheels to clamp seeds, but as the center distance of the wheels continues to shrink, the impact and bounce effect on the seeds will grow. The rotating speed of the seed disk was set to a low speed, and the rotating speed of the seed disk was established to be 20 r/min, in order to reduce the impact of the rotating speed of the seed metering device on the average stress of the seeds in the feeder wheel clamp. The range of the wheels' center distance between the primary and the auxiliary feeder wheel was found to be 35–39 mm, and finally, the single factor experiment was determined with the wheels' center distances of 35, 37, and 39 mm. This was carried out in order to reduce the collision between the feeder wheel and the seed, while also taking into account the feeder wheel's effect on seed receiving. The line graph of the stress on corn seeds over time is shown in Figure 12.

The average and peak values of the stress on corn seeds under various wheel center distances exhibited a significant difference, as shown in Figure 12. The average stress was 1.84 N and the peak stress was 7.13 N at 0.68 s when the wheels' center distance was 35 mm (Figure 12a). The corresponding average stress was 2.30 N, and the peak stress was 5.51 N at 0.64 s when the wheels' centers were 37 mm apart (Figure 12b). The average stress was 1.84 N and the peak stress was 7.13 N at 0.68 s when the wheels' center distance was 35 mm (Figure 12c). Through numerical comparison, it can be seen that the average value of the feeder wheel's stress on the seeds decreased by 0.46 N and the peak value rose by 1.62 N when the wheels' center distance was raised from 35 mm to 37 mm. This shows that the wheels' center distance of 35 mm could only increase the stress on the seeds at a

certain moment, but the average stress on the seeds in the whole process of running was still small, which was smaller than that of the feeder wheel at 37 mm. The average value of the feeder wheel's stress on the seeds was reduced by 1.34 N and the peak value was reduced by 2.08 N when the wheels' center distance was increased from 37 mm to 39 mm. This indicates that the larger wheels' center distance reduced the feeder wheel's stress on the seeds, and the clamping performance suffered as a result.

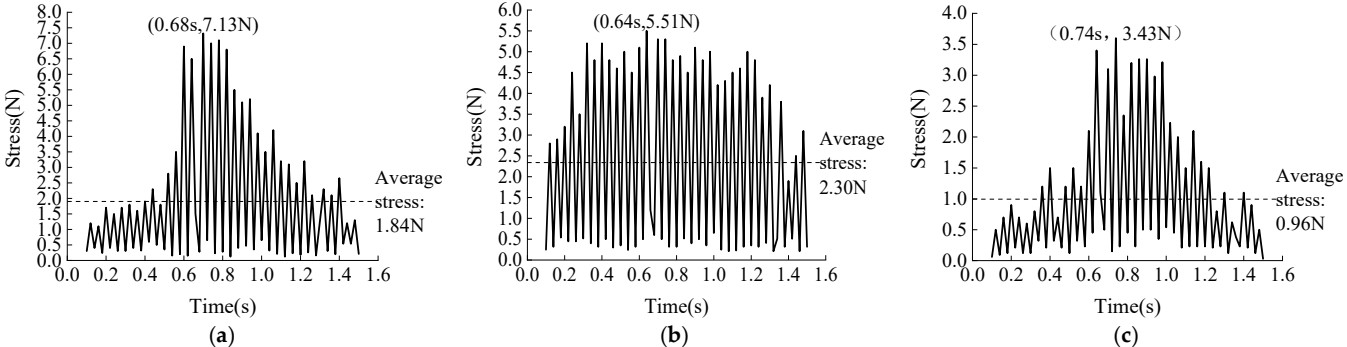

**Figure 12.** Effect of the different wheels' center distance on the stress of corn seeds. (**a**) Line graph of the seed stress change when the wheels' center distance was 35 mm. (**b**) Line graph of the seed stress change when the wheels' center distance was 37 mm. (**c**) Line graph of the seed stress change when the wheels' center distance was 39 mm.

### 3.2. Analysis of the Seed Cavity's Seed Entering Speed

This study was entirely expressed by the speed and finger length of the primary feeder wheel, with the finger length ratio of the primary feeder wheel and the auxiliary feeder wheel being 3:1 and the speed ratio being 1:1.7. This was conducted to make the following description of the speed and finger length of the feeder wheel easier to understand.

#### 3.2.1. Analysis of Feeder Wheel Speed on the Speed of Seeds Entering the Seed Cavity

The wheels' center distance was set to 37 mm, and the finger length was set at 12 mm, in order to examine how corn seeds changed in the seed cavity as the feeder wheel speed was varied. It was decided that the feeder wheel speed be changed from 500 to 620 r/min, so the feeder wheel speeds were 500, 560, and 620 r/min, respectively, to meet the needs of the actual sowing operation and account for the clamping effect of the feeder wheels on seeds. Figure 13 illustrates the changing trend of the speed of corn seeds entering the seed cavity at various rotational speeds by setting up a speed monitoring block in the seed cavity and recording the seed speed passing through the block at any time.

As seen in Figure 13, there was a general upward trend in the speed of corn seeds entering the seed cavity as the feeder wheel speed increased. When the feeder wheel speed was 500 r/min, the broken line of seed speeds changed greatly, indicating that the speed of seed entering the seed cavity fluctuated greatly at this time; this speed fluctuation was further reduced when the feeder wheel speed was increased to 560 r/min, and this speed fluctuation tended to stabilize when the feeder wheel speed increased to 620 r/min. The analysis of the aforementioned data revealed that although the speed fluctuated significantly and the overall stability of seed guiding declined, the slower the feeder wheel speed, the lighter the collision between the seeds and the wall of the seed cavity, and the slower the speed of the seeds entering the seed cavity. The speed of seeds entering the seed cavity and the stability of seed guiding increased with the feeder wheel speed, but as seeds enter the cavity at a faster rate and collide more violently with its wall, the phenomenon of seeds escaping from the cavity will result. The study above leads to the conclusion that the seed can enter the seed cavity at a good speed and be maintained at a pace when the feeder wheel speed is 560 r/min. In order to improve the likelihood of seeds entering the cavity steadily, the speed at which corn seeds enter the seed cavity can be changed by changing

the feeder wheel speed. This provides a basis for figuring out the operating parameters of the seed receiving system.

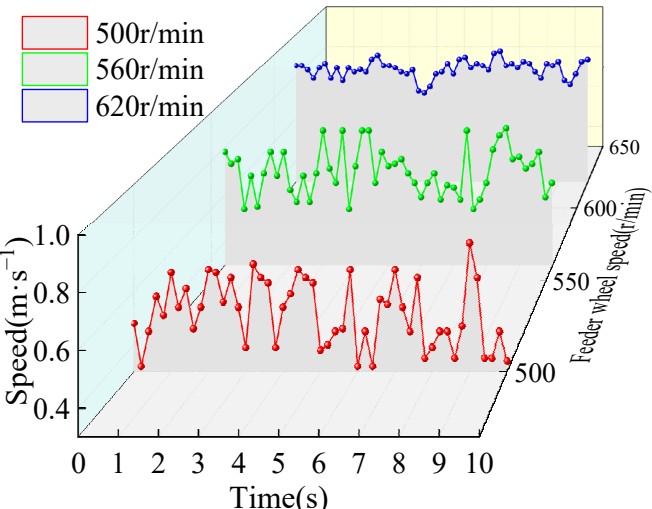

**Figure 13.** Effect of different feeder wheel speeds on the speed of corn seeds entering the seed cavity.

3.2.2. Analysis of Finger Length on the Speed of Seed Entering Seed Cavity

The wheels' center distance was set to 37 mm, and the feeder wheel speed was set to 560 r/min, in order to examine how corn seeds changed in the seed cavity as the finger lengths varied. Finger lengths of 10, 12, and 14 mm were used in the simulation experiments because that was the range of finger lengths that was found to be necessary to make the fingers have the effect of clamping seeds. Figure 14 illustrates the changing trend of the speed of corn seeds entering the seed cavity at various finger lengths by setting up a speed monitoring block in the seed cavity and recording the seed speed passing through the block at any time.

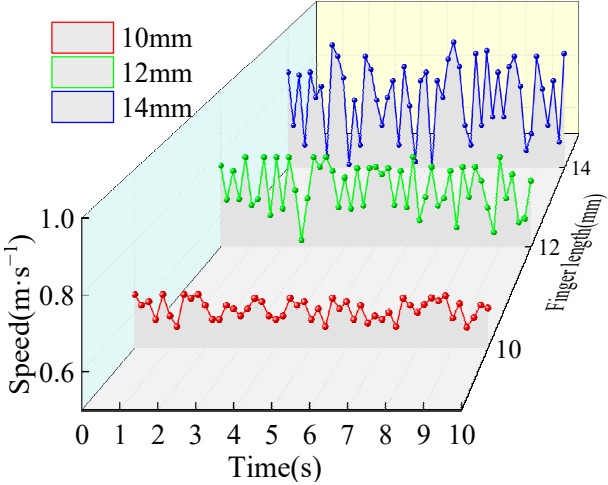

**Figure 14.** Effect of different finger lengths on the speed of corn seeds entering the seed cavity.

Figure 14 illustrates how the rate at which corn seeds enter the seed cavity generally increases as the finger lengthens. When the finger length was 10 mm, the change amplitude of the speed broken line of seeds was small, which indicates that the speed fluctuation of seeds entering the seed cavity was minimal; when the finger length increased to 12 mm, the speed fluctuation of seeds entering the seed cavity was increased; when the finger length increased to 14 mm, the speed fluctuation of seeds entering the seed cavity was even more increased, indicating an extremely unstable state. By examining the change

law of the aforementioned data, it was discovered that reducing the finger length can improve the accuracy of the seed conveying belt by stabilizing the speed of seeds entering the seed cavity. The speed at which seeds enter the seed cavity is, however, relatively slow, which can lessen seed-to-cavity wall collisions but is not appropriate for high-speed seed guiding operations. The speed at which seeds enter the seed cavity can be accelerated by lengthening the finger, but this increases the risk of the high-speed seed guiding operation becoming unstable due to the large fluctuations in the seed entry speed. According to the above study, when the finger length is 12 mm, the seed can enter the seed cavity at a good speed and be maintained at a steady pace. Therefore, by adjusting the finger length, it is possible to alter the rate at which corn seeds enter the seed cavity and achieve a good rate of entry, improving the stability of seed transportation on the seed conveying belt.

### 3.3. Verification Test

The wheels' center distance, feeder wheel speed, and finger length of the seed receiving system were used as test factors, and the seed receiving rate of the seed receiving system and the variation coefficient of the seed cavity spacing of the seed conveying belt was used as the performance evaluation indices to verify the simulation results. The partial shell, seed disk, and rubber sealing ring of the seed metering device were printed in 3D transparently to intuitively reflect the movement state of the feeder wheel. The displacement curve of corn seeds was recorded using a high-speed camera, and the smoothness and fluctuation of the seed displacement curve were used to calculate the seed receiving rate. In order to observe the distribution of seeds entering the seed cavity through the feeder wheel, the shell of the seed conveying belt was treated to be transparent. The accuracy of seeds entering the seed cavity was measured by computing the variation coefficient of the seed cavity spacing for each group of test seeds.

The test site was Heilongjiang Bayi Agricultural University's sowing lab. According to Figure 15, the test apparatus was primarily made up of a JPS-16 computer vision seed metering test bed (the maximum sowing speed was 16 km/h), a PCO.dimaxCS3 high-speed camera (German PCO.dimax CS high-speed camera, Nikon lens, image shooting program was Camware), an air-suction corn seed metering device, and a belt-type high-speed corn seed guiding device.

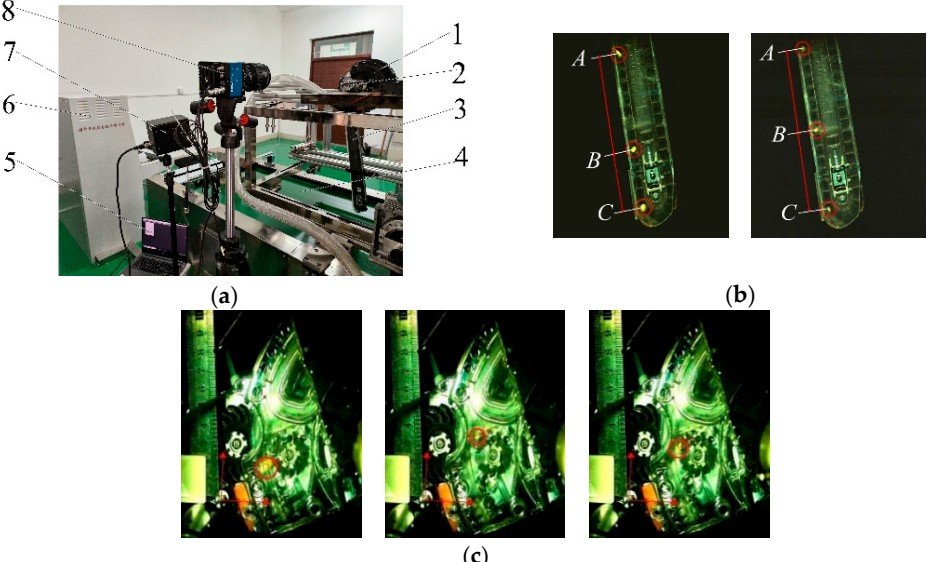

**Figure 15.** Test equipment for the performance detection of the seed receiving system. 1. Air-suction corn seed metering device. 2. Seed metering drive motor. 3. Belt-type high-speed corn seed guiding device. 4. Seed bed belt. 5. Computer. 6. Seed test controller. 7. LED lighting. 8. High-speed camera. (**a**) JPS-16 computer vision seed metering test bed. (**b**) Distribution of seeds in seed conveying belt. (**c**) Seed receiving process under high-speed camera.

Each group was repeated three times, and the average of the statistical processing data was taken as the test result, as shown in Figure 16.

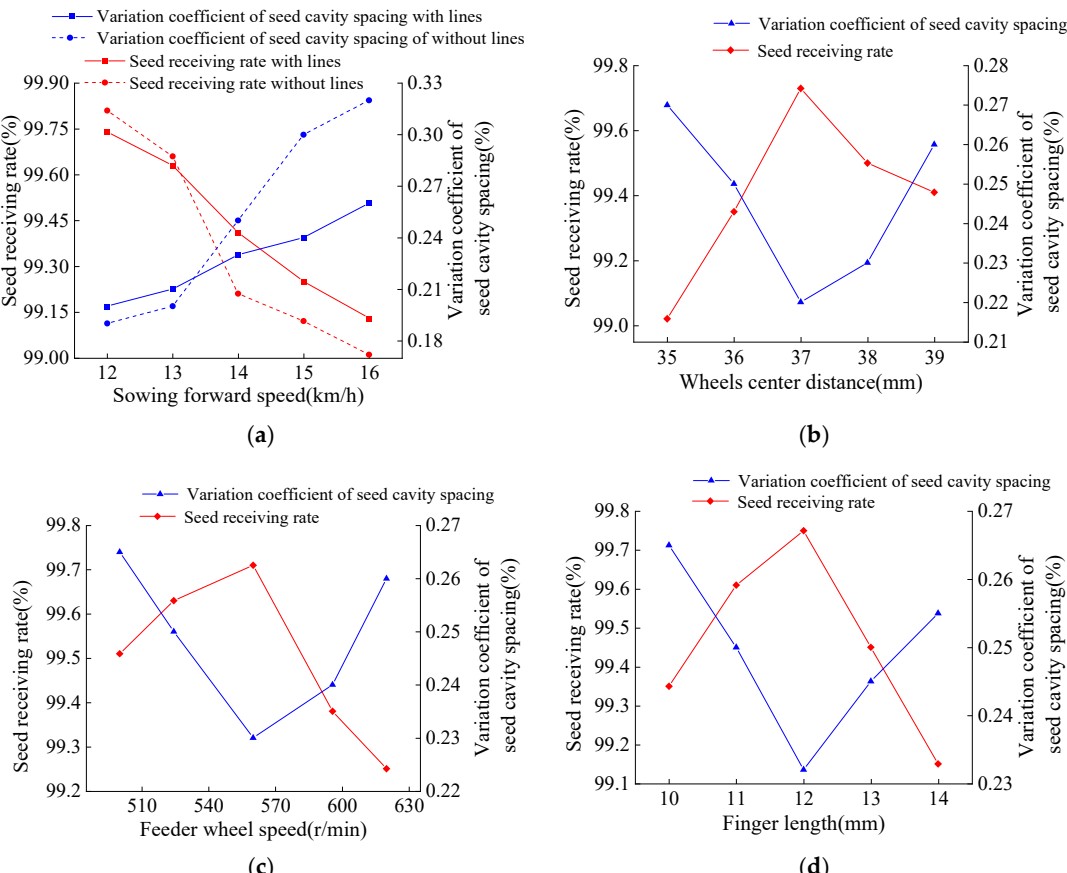

**Figure 16.** Results of the seed receiving rate and variation coefficient of the seed cavity spacing under bench test. (**a**) Comparing experimental results with/without herringbone lines; (**b**) Changes of test index with wheels' center distance; (**c**) Changes of test index with the feeder wheel speed; (**d**) Changes in the test index with finger length.

The seed receiving rate and the variation coefficient of the seed cavity spacing between the feeder wheel with herringbone lines and without herringbone lines at low sowing speeds were not visibly different, as shown in Figure 16a. The seed receiving rate and the variation coefficient of the seed cavity spacing of the feeder wheel with herringbone lines were clearly superior to those of the feeder wheel without herringbone lines when the sowing speed was between 13 and 16 km/h, and they were essentially the same as the variation law of the stress of the feeder wheel with herringbone lines in the simulation.

As can be seen from Figure 16b, with the increase in the wheels' center distance, the seed receiving rate first increased and then decreased, and the variation coefficient of the seed cavity spacing first decreased and then increased, which was the same as the variation law of stress on a feeder wheel with different wheel center distances in the simulation.

Figure 16c,d shows that the seed receiving rate increased and then decreased as the feeder wheel speed and finger length increased, while the variation coefficient of the seed cavity spacing first decreased and then increased, following the same variation law as the feeder wheel with various feeder wheel speeds and finger lengths in the simulation.

## 4. Conclusions

(1) Taking the belt-type high-speed corn seed guiding device with a seed receiving system as the research object, the seed receiving system was analyzed, the seed dynamics model in the process of clamping, transporting, and releasing seeds by feeder wheel

was established, the improved method of adding herringbone lines on the finger surface was put forward, and it was clear that the main factors affecting the seed receiving stability and the accuracy of seeds entering the seed cavity were the wheels' center distance, the feeder wheel speed, and the finger length.

(2) By using EDEM simulation technology, the seed receiving process was simulated. The findings demonstrate that by adding herringbone lines to the feeder wheel, the stress on seeds can be clearly increased. The average value of the stress on the seeds was the highest at a wheel center distance of 37 mm. The stability and speed fluctuation of seeds introduced into the seed cavity were better when the feeder wheel speed was 560 r/min. The speed of fluctuation and stability of seeds introduced into the seed cavity were better when the finger length was 12 mm.

(3) The bench test results were largely consistent with the virtual simulation according to the results of the verification test, so it can be used to model and examine how the seed receiving system of a belt-type high-speed corn seed guiding device functions.

**Author Contributions:** Conceptualization, C.M. and S.Y.; Data curation, C.M., Y.L., S.W. and G.W.; Formal analysis, C.M., G.T. and F.G.; Funding acquisition, G.W. and F.G.; Investigation, C.M. and S.Y.; Methodology, C.M. and S.Y.; Project administration, Y.L. and S.W.; Resources, S.Y. and Y.L.; Software, C.M.; Supervision, S.Y. and Y.L.; Validation, C.M., S.Y., G.T. and Y.L.; Visualization, C.M., S.W., G.W. and F.G.; Writing—original draft, C.M. and S.Y.; Writing—review and editing, C.M. and G.T. All authors have read and agreed to the published version of the manuscript.

**Funding:** This research was funded by the National Natural Science Foundation, grant number 52275246, the Key research and development plan of Heilongjiang Province—major project topic, grant number 2022ZX05B02-02, and the Innovative scientific research project for postgraduates of Heilongjiang Bayi Agricultural University, grant number YJSCX2022-Z02.

**Institutional Review Board Statement:** Not applicable.

**Data Availability Statement:** The data will be available through contacting the corresponding author.

**Conflicts of Interest:** The authors declare no conflict of interest.

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
