# Peer review of "Research on Receiving Seeds Performance of Belt-Type High-Speed Corn Seed Guiding Device Based on Discrete Element Method"

_agriculture, doi:10.3390/agriculture13051085_

Round 1
Reviewer 1 Report
The manuscript titled’ ‘Research on Receiving Seeds Performance of Belt-Type High-Speed Corn Seed Guiding Device Based on Discrete Element Method’ is a very interesting research paper. There is an evident scientific justification for the publication of this scientific work. The methodological concept of the complete manuscript is at a high scientific level (except for the references). I am happy to recommend his publication. However, some parts of the manuscript need to be revised and supplemented. I suggest the following minor revisions to the manuscript:
1. The importance and thus significance of work has not been properly justified in the background introduction.
2. Please give the source of DEM parameters, considering the relative accurate parameters is the foundation of DEM feasibility
3. Please modify the Figure 13 and Figure 14 due to the text disorder.
Quality of english language is well.
Author Response
Dear reviewer and editor,
First of all, I would like to express my gratitude for the help and support you provided in reviewing my article. Thank you for taking the time out of your busy schedule to carefully review my work and provide me with valuable feedback and suggestions. Your professional guidance and advice have been invaluable in improving and refining my manuscript, and have provided important insights for our research.
My team and I have carefully considered your feedback and made the necessary revisions and modifications. We recognize the importance of your comments in guiding and shaping our research, and have highlighted the revised sections in red in the revised version. We hope to have your approval on the changes made.
Responses to reviewer (original comments by reviewer are in blue color)
1.Comment: The importance and thus significance of work has not been properly justified in the background introduction.
1.Reply: Thank you for your advice. The importance and significance of this study have been added at the end of the first paragraph of the "Introduction".
2.Comment: Please give the source of DEM parameters, considering the relative accurate parameters is the foundation of DEM feasibility.
2.Reply: As per your advice, I provided necessary references as the data source for simulation in the chapter "2.4.3. Simulation Parameter Setting."
3.Comment: Please modify the Figure 13 and Figure 14 due to the text disorder.
3.Reply: Thank you for your wise counsel. Regarding the issues in Figures 13 and 14, we believe the waterfall chart is appropriate for the following reasons:
(1) The waterfall chart is a form of chart that can clearly display the factors or process of data change, as well as the cumulative summary when adding or deleting values, reflecting the degree and results of data in different times or affected by different variables.
(2) The waterfall chart arranges the data in the chart so that it appears to be a waterfall hanging, which boosts the visual effect and appeal of the chart and also helps readers grasp the influence of factor changes on seed speed more intuitively.
As a result, we believe that using a waterfall chart can help readers grasp the impact of factor modification on seed speed in a more intuitive way, which is information that other chart types cannot simply portray. We hope that the reviewer will understand and appreciate our choice, and that our essay will receive greater support and validation as a result. thank you very much.
Thank you again for your support and assistance, and I wish you all the best in your work and good health!
Sincerely,
Chengcheng Ma

Reviewer 2 Report
This paper aims to solve the problem of seeds colliding with each other and rebounding from the inner wall of the seed duct during high-speed corn sowing, resulting in poor sowing quality. The main factors affecting the stability of seed reception and the accuracy of seed entry into the seed cavity are studied, and a dynamic model for seed clamping, transportation, and release is established, A belt type high-speed corn seed guidance device with a herringbone pattern added to the surface of the finger was proposed to solve this problem. EDEM software was used to simulate and analyze the effects of three main factors: finger length, feed wheel speed, and center distance between the main and auxiliary wheels. A bench experiment was built to verify the simulation results. Provide theoretical basis for the optimization experiment of the belt type high-speed corn seed guide device.
The research method of this paper is reasonable and the writing format is relatively standard, but there are the following problems:
(1) When using a feeder wheel with a herringbone pattern in 3.1.1, the stress on the corn seed is relatively high, while the statement mentioned in the final summary of the article is that the stress on the corn seed decreases after using the herringbone pattern, which is contradictory.
(2) In this paper, we only explored the addition of herringbone patterns on the fingers, and in future research, we can add control groups with different patterns to seek the optimal solution.
(3) In this paper, the optimal values of finger length, feed wheel speed, and center distance between main and auxiliary wheels for seed guidance were obtained through EDEM software simulation. When multiple factors change at the same time, the parameter values corresponding to the best guidance effect may vary compared to single factors.
English is very good.
Author Response
Dear reviewer and editor,
First of all, I would like to express my gratitude for the help and support you provided in reviewing my article. Thank you for taking the time out of your busy schedule to carefully review my work and provide me with valuable feedback and suggestions. Your professional guidance and advice have been invaluable in improving and refining my manuscript, and have provided important insights for our research.
My team and I have carefully considered your feedback and made the necessary revisions and modifications. We recognize the importance of your comments in guiding and shaping our research, and have highlighted the revised sections in red in the revised version. We hope to have your approval on the changes made.
Responses to reviewer (original comments by reviewer are in blue color)
1.Comment: When using a feeder wheel with a herringbone pattern in 3.1.1, the stress on the corn seed is relatively high, while the statement mentioned in the final summary of the article is that the stress on the corn seed decreases after using the herringbone pattern, which is contradictory.
1.Reply: Thank you for your care. Your review helped me find a mistake. I mistyped "increased" as "reduced" in the conclusion due to carelessness, and I have changed it.
2.Comment: In this paper, we only explored the addition of herringbone patterns on the fingers, and in future research, we can add control groups with different patterns to seek the optimal solution.
2.Reply: Thank you very much for your reminder and encouragement. Your suggestion has provided valuable inspiration for my later research. I deeply admire your care and patience, and I feel very lucky to get your guidance. Thank you again for your help and support!
3.Comment: In this paper, the optimal values of finger length, feed wheel speed, and center distance between main and auxiliary wheels for seed guidance were obtained through EDEM software simulation. When multiple factors change at the same time, the parameter values corresponding to the best guidance effect may vary compared to single factors.
3.Reply: I completely agree with you. Although the multi-factor experiment may completely explain the interaction between variables, we primarily employ the single-factor experiment approach in this work to investigate the influence of variables on seed speed and stress. Thus, the degree of influence of each variable may be determined more precisely, and potential data mixing and inaccuracies in multi-factor trials can be avoided. Of course, we recognize the value of multi-factor experiments, but given the workload and data accuracy of EDEM simulation, we chose single-factor experiments as the primary strategy for this study. Thank you once more for your help and interest in our research!
Thank you again for your support and assistance, and I wish you all the best in your work and good health!
Sincerely,
Chengcheng Ma

Reviewer 3 Report
1.At Figure 14. Effect of different finger lengths on the speed of corn seeds entering seed cavity. What is the value of rpm.
2.Please give more details why the stability and speed fluctuation of seeds introduced into the seed cavity are better when the feeder wheel speed is 560 r/min. The speed of fluctuation and stability of seeds introduced into the seed cavity are better when the finger length is 12mm.
3. On Figure 16. Can you justify why seed receiving rate was declined rapidely after after Feeder wheel speed started to be 570 rpm and also after Finger length changed to 12 (mm).
Can you give details about EDEM software to simulate and optimize the seed placement process in precision planters?
Author Response
Dear reviewer and editor,
First of all, I would like to express my gratitude for the help and support you provided in reviewing my article. Thank you for taking the time out of your busy schedule to carefully review my work and provide me with valuable feedback and suggestions. Your professional guidance and advice have been invaluable in improving and refining my manuscript, and have provided important insights for our research.
My team and I have carefully considered your feedback and made the necessary revisions and modifications. We recognize the importance of your comments in guiding and shaping our research, and have highlighted the revised sections in red in the revised version. We hope to have your approval on the changes made.
Responses to reviewer (original comments by reviewer are in blue color)
1.Comment: At Figure 14. Effect of different finger lengths on the speed of corn seeds entering seed cavity. What is the value of rpm.
1.Reply: Thank you for your comment. Maybe my incorrect description of the speed unit influenced your review. The speed of the feeder wheel at this moment is 560 r/min, as specified in the second line of the first paragraph of the section "3.2.2. Analysis of finger length on the speed of seed entering seed cavity," and I have adjusted the expression of the speed unit.
2.Comment: Please give more details why the stability and speed fluctuation of seeds introduced into the seed cavity are better when the feeder wheel speed is 560 r/min. The speed of fluctuation and stability of seeds introduced into the seed cavity are better when the finger length is 12mm.
2.Reply: Thank you for your advice. I've added a description of the broken speed line to the speed analysis and summarized it so that it corresponds better to the conclusion.
3.Comment: On Figure 16. Can you justify why seed receiving rate was declined rapidly after Feeder wheel speed started to be 570 rpm and also after Finger length changed to 12 (mm).
Can you give details about EDEM software to simulate and optimize the seed placement process in precision planters?
3.Reply: Thank you very much for raising this question. The trend of data changes was obtained through high-speed camera experiments, and we have not analyzed the specific reasons yet. However, your question reminds us of the focus of our next stage of research. We will try to explore the reasons for this pattern in the next round of experiments. Thank you very much for your feedback, which is very helpful to our research.
The basic operation process of seeding simulation through EDEM is as follows:
- Model creation: Use the modeling tools in EDEM software to create a 3D model of a precision seeder, including components such as seed hoppers, conveyors, vibratory trays, rotating disks, and seeders.
- Set physical parameters: To accurately simulate the seeding process, set the appropriate physical parameters for each component, such as weight, shape, friction coefficient, elastic modulus, etc. These parameters can be set in the EDEM software.
- Set initial conditions: Set the initial conditions, including the number, position, and velocity of seeds, as well as the initial state of the machine, such as vibration intensity, rotation speed, etc.
- Run simulation: Run the simulation program in the EDEM software to simulate the operation of the seeder. During the simulation, EDEM calculates the motion trajectories, velocities, accelerations, collisions, and other information of each particle, and updates the model's status in real-time.
- Analyze results: Analyze the simulation results, including the distribution, density, depth, and other information of seeds, as well as the efficiency and quality of the operation of the machine. Through the analysis of results, optimize design parameters to improve the seeding efficiency and planting quality.
Thank you again for your support and assistance, and I wish you all the best in your work and good health!
Sincerely,
Chengcheng Ma
